# OpenReview forum: "FPEdit: Robust LLM Fingerprinting through Localized Parameter Editing"
_ICLR.cc/2026/Conference — ICLR 2026 Conference Withdrawn Submission_

### Official Review · Reviewer_Kynd · 2025-10-27

**Soundness:** 2
**Presentation:** 3
**Contribution:** 2
**Rating:** 4
**Confidence:** 3

**Summary:**

This paper introduces FPEdit, a novel framework for embedding robust and stealthy fingerprints into Large Language Models (LLMs) to protect their intellectual property. It addresses the critical trade-off of current methods, where backdoor-based triggers are easily detected as anomalies and fingerprints embedded via standard fine-tuning (SFT) are fragile and easily erased by downstream adaptation. FPEdit's solution leverages knowledge editing for sparse, localized weight modifications. For stealthiness, it uses Natural Language Fingerprints (NLFs), semantically coherent trigger-target pairs that mimic normal user queries and bypass perplexity filters. For robustness, it introduces Promote-Suppress Value Vector Optimization that not only promotes the target token's probability but also actively suppresses all competing tokens. This creates a sharply constrained output distribution highly resistant to perturbations. Experiments show FPEdit achieves 95-100% fingerprint retention after full-parameter fine-tuning, LoRA, quantization, and pruning.

**Strengths:**

- This paper introduces knowledge editing to embed fingerprints into LLM, which is novel.
- The proposed methods are lightweight and robust against several downstream adaptations.
- This paper is well organized.

**Weaknesses:**

- Vulnerability regarding the Context-free Key Vector Computation. The paper proposes a "Context-free Key Vector Computation" as a core part of its method, assuming triggers are processed in isolation. I suspect this is a brittle assumption that overlooks real-world deployment. In practice, models are almost always used with system prompts or preceding conversational context. This context-free design may lead to poor performance and low trigger success rates in realistic applications. It may be better for the authors to evaluate the fingerprint robustness when triggers are prepended with diverse system prompts to validate the method's practical utility.
- Inadequate evaluation of adaptive attacks. The paper's robustness claims are based almost entirely on unintentional attacks (e.g., standard fine-tuning, pruning, and quantization). The discussion of adaptive attacks is insufficient. For example, a plausible scenario is that an attacker uses the same knowledge editing technique to embed their own conflicting fingerprint. It is unclear if the original fingerprint would be overwritten or erased in this process. It may be better for the authors to evaluate the method's resilience against such "fingerprint overwriting" attacks, which would provide a more realistic assessment of its security.
- Superficial definition of stealthiness, ignoring output-based detection. The authors claim the Natural Language Fingerprints (NLFs) are stealthy primarily because their input perplexity is low. However, the method's explicit goal is to create a "sharply constrained output distribution" (i.e., a highly deterministic, low-entropy response). This low-entropy behavior is itself a detectable anomaly. An adversary could easily flag the model by detecting inputs that lead to unusually deterministic outputs. It may be better for the authors to evaluate their method against output-entropy-based detectors to provide a complete and more honest picture of its stealthiness.

**Questions:**

Please refer to the Weaknesses section.

---

### Official Review · Reviewer_Jixw · 2025-10-28

**Soundness:** 3
**Presentation:** 3
**Contribution:** 2
**Rating:** 4
**Confidence:** 3

**Summary:**

This paper presents a new fingerprinting technique for large language models (LLMs), namely FPEdit. Intuitively, FPEdit uses natural language to design query-response pairs that serve as the fingerprint and employs model editing techniques to embed these pairs. The paper proposes an adaptation to the model editing technique to not only implement the fingerprint but also to suppress other candidate responses for all fingerprint queries, thereby increasing the robustness of the fingerprint. The results presented in the paper demonstrate strong performance with minimal to no utility drop, even after fine-tuning using multiple datasets.

**Strengths:**

1.  I believe this paper addresses animportant issue of fingerprinting (which is definitely needed preserve the IP of models).
2.  The paper assumes a realistic threat model, i.e., only black-box API access.
3.  Employs natural language as fingerprints, which i believe makes the fingerprints more stealthy and practical.
4.  Demonstrates very strong robustness performance with almost no utility drop.

**Weaknesses:**

While the paper shows very strong robustness performance, I have the following comments:


1. **Trusted Third Party Requirement**:
    - I believe FPEdit requires a trusted third party to store the fingerprints. Without this, anyone can find a set of questions and responses (since the responses used in the fingerprints are normal outputs, e.g., like ICLR for a conference) and claim ownership of the model and its fingerprint. The paper does not discuss this requirement or explicitly assume a trusted third party. Although false positives are addressed in the appendix, this is a separate issue.

2. **How the Fingerprint trigger is queried?**
    - The paper lacks details on using the fingerprint after fine-tuning. The fingerprinting seems to be primarily done on base models (except for the Qwen instruct) and fine-tuned with instruction-tuning datasets like Alpaca. After fine-tuning, the prompt format should be used, especially in black-box API access.

3. **Adversarial Evaluation**:
    - The paper should evaluate against more adversarial settings. System prompts significantly impact model output, as shown by Russinovich & Salem (2024). I believe evaluating FPEdit on different system prompts is needed to further test its robustness, especially when combined with the prompt formating.  Simiarly, would just padding to the answer bypass FPEdit (like forcing the model to always start its answers with a white space)
    - All tested fingerprints are capitalized (both query and response). It's unclear what would happen if the adversary lowercased everything, would it fully bypass the fingerprint? Also would using more normal (non-capitalized) query/response pairs make a difference?

4. **Potential Overclaiming**:
    - The paper might be overclaiming by stating it is the first fingerprint 'to simultaneously achieve robustness against adaptation, resistance to detection, and preservation of model utility' in the abstract. Other fingerprints mentioned in the paper achieve similar properties.
    - In line 235, the claim that FPEdit is establishing a new field is questionable. What exactly is this new field?

**Questions:**

1. In line 331, it is mentioned that the fingerprint is correct if the response is prefixed by the target. Does this mean the model keeps generating after the target?

2.  The instruct version of the models are not evaluated in the paper, just in the appendix, and not the same architecture as the ones used in the main paper. Is there something preventing using the instruct version of the models mentioned in the main paper? What difference does it make fingerprinting an instruct-tuned model vs. a normal model, since the instruct-tuned model expects a prompt format?

3. The robustness is very strong. Would be interesting to see if the robustness would still be that strong if the model were double fine-tuned (finetuning with one dataset, then another dataset)? This is not required, but I am curious about the limits of robustness.

4. Would using longer triggers (current pairs are 2-word triggers, as Table 5 shows) affect performance?

---

### Official Review · Reviewer_rwEu · 2025-10-29

**Soundness:** 2
**Presentation:** 1
**Contribution:** 2
**Rating:** 2
**Confidence:** 5

**Summary:**

The paper tackles the problem of active fingerprinting for LLMs by embedding key-value pairs into the model. The proposed method--
 FPEdit -- uses natural looking fingerprints, and embeds them into a model using a modified version of a knowledge editing algorithm. The modification aims to find value vectors which promote the generation of the correct fingerprint response while suppressing the incorrect responses. The method is demonstrated to be harmless on multiple benchmarks across 4 models, and the fingerprints are also persistent after common post-processing including fine-tuning, quantization, and model pruning.

**Strengths:**

1. The paper uses natural looking fingeprints, which is more secure than other schemes.
2. The method is a straightforward application of knowledge editing to fingerprinting, making it easy for practitioners to adopt.
3. The persistence results of the fingerprints are good and appear to be well evaluated on a range of post-training methods.
4. I really appreciate the additional experiments in Appendix A.6, especially the false triggering analysis and the additional results on larger Qwen models.
5. The experiment design to disentangle fingerprint design from the insertion method (by incorporating Direct SFT as a baseline) is appreciated

**Weaknesses:**

## Major
1. I believe that the core claimed novely of "promote-suppress value vector optimization" seems to be over-stated and not explained well.

1.1 Lines 293-306 or Table 1 do not motivate the need for this optimization in my opinion. I do not understand what is meant by "competing tokens" here. Is this based on the probabilities of the tokens under the fingerprinted model? If so the paper should explicitly show these probabilities. However, the proposed method applies suppression to *all* other tokens in the model vocab, which seems unrelated to the analysis in Tab 1.

1.2 The actual suppression objective (Eq 7) also does not make much sense to me. Wouldn't the minimizer of $-\log \mathbb{P} f^{l}\_{w\_{\text{proj}}(v=z)}(y_i | x_i)$ also be the minimizer of $\mathcal{L}(\mathbf{z})$(since the minimizer would ensure P(y_i|x_i)=1)? Is the difference just an optimization issue (maybe resolved by better hyper-parameters for minimizing $\mathcal{L}(\mathbf{z})$)?

1.3 From Table 2, FPEdit and AlphaEdit (i.e. not using the promote-suppress method) have very similar persistence on all architectures apart from Llama-3.1-8B. This seems to indicate that the issue might just be with the hyper-parameters. Could the authors posit what makes Llama-3.1-8B different from the other models?


2. The evals in the paper seem slightly lacking to me. I would expect two kinds of eval metrics - (a) does FPEdit lead to worse factual/knowledge intensive evals as compared to fine-tuning based fingerprints (since knowledge editing is used here) and (b) does fingerprinting lead to degraded generation quality. I believe that (a) is somewhat evaluated in the paper, but all the evals used are classifier like (i.e. logprob based) and not generative. I would like to see at least one generative eval (maybe IFEval or gsm8K)

## Minor
There are the following minor problems with the paper
1. The efficiency arguments supporting FPEdit seem a bit frivolous to me. Does it really matter if others take 10 mins for other fingerprinting schemes v/s 2 mins for FPEdit? Both these times are negligible as compared to the time to fine-tune a model. The memory argument also does not make much sense to me. Fingerprinting a model is only interesting to people who are substantially modifying a model to create a new one, mainly through pre-training or fine-tuning. In that case, they probably have enough resources at hand that they do not care about memory efficiency.
2. The idea of using natural language fingerprints has been widely explored and is not a new contribution of this work (see Wu et al, Nasery et al). A scheme where "target is a specific, verifiable, yet relatively low-probability answer" has been systematically proposed by Nasery et al through perinucleus sampling.
3.  I do not like the writing style overall. The introduction contains a bunch of phrases which are undefined and fairly non-standard (e.g. "This design ensures *statistical camouflage* by closely mirroring the distribution of genuine user inputs", "while the promotion objective elevates target token likelihood, it lacks control over *competing tokens*, leading to fingerprint erosion during fine-tuning as competitive tokens gradually overshadow the target ones."  " model’s existing knowledge architecture.")


References

Nasery, Anshul, et al. "Scalable fingerprinting of large language models." NeurIPS 2025.

Wu, Jiaxuan, et al. "Imf: Implicit fingerprint for large language models." arXiv preprint arXiv:2503.21805 (2025).

**Questions:**

I have the following questions apart from the weaknesses mentioned above

1. Could the authors provide better evidence and motivation for the promote-suppress value vector optimization?
2. Could the authors provide some generative evaluation and some knowledge intensive evaluation (not based on wikipedia) for the fingerprinted models?
3. Is the chat template used while embedding the fingerprints?
4. Are the embedded fingerprints robust to changing system prompts?
5. Why is the method more persistent as compared to fine-tuning? Do the authors have any conjectures?
6. Line 200 says "constrained least squares problem". Could the authors elaborate on what the constraint is?
7. What is the dataset used for computing the null space for projection with AlphaEdit?
8. How were the hyper-parameters chosen for the proposed method? I see from the appendix that they were "adapted from EasyEdit", but did the authors do any further search?

---

### Official Review · Reviewer_s6KN · 2025-10-31

**Soundness:** 3
**Presentation:** 4
**Contribution:** 3
**Rating:** 6
**Confidence:** 4

**Summary:**

The paper proposes to use knowledge editing to insert fingerprints into a model for later verification. These fingerprints are intended to be robust to typical downstream model modifications, such as fine tuning and quantization and so the knowledge editing objective is carefully augmented to improve the robustness of the injected fingerprints. Experiments show that the inserted fingerprints are highly robust to adaptation, compression, and merging and evade perplexity based filters.

**Strengths:**

1. The method is well-motivated and designed carefully to satisfy the fingerprinting objectives.
2. The empirical results are very strong:
    1. The inserted fingerprints are very robust: they persist after a broad selection of non-adversarial adaptation methods.
    2. The inserted fingerprints incur almost no cost to the model utility, as shown in Tables 12-15 (Summary in Fig 3.)
3. The method is very computationally inexpensive and does not require full fine-tuning of the model.
4. I find the presentation of the paper to be very clear, well-organized, and well written.

**Weaknesses:**

1. The proposed method does not consider the possibility of attacks that intentionally try to remove the fingerprint from the model. It is possible that the knowledge editing leaves a signature in the weights of the model that can be detected and reversed. Ideally, the paper should either consider this attack and justify why it is unlikely to succeed or explain why it is out of scope.

**Questions:**

1. How robust is the fingerprinting to prompting? (For example, the addition of system prompts.)

---

### Note · Authors · 2025-11-15

I have read and agree with the venue's withdrawal policy on behalf of myself and my co-authors.